# Versatile bubble maneuvering on photopyroelectric slippery surfaces

Haiyang Zhan[1], Zichao Yuan[1], Yu Li[1], Liang Zhang[1], Hui Liang[2], Yuhui Zhao[2,3], Zhiguo Wang[2,3], Lei Zhao[1], Shile Feng[1] & Yahua Liu [1] ✉

Contactless bubble manipulation with a high spatiotemporal resolution brings a qualitative leap forward in a variety of applications. Despite considerable advances, light-induced bubble maneuvering remains challenging in terms of robust transportation, splitting and detachment. Here, a photopyroelectric slippery surface (PESS) with a sandwich structure is constructed to achieve the versatile bubble manipulation. Due to the generated dielectric wetting and nonuniform electric field under the irradiation of near infrared (NIR) light, a bubble is subject to both the Laplace force and dielectrophoresis force, enabling a high-efficiency bubble steering. We demonstrate that the splitting, merging and detachment of underwater bubbles can be achieved with high flexibility and precision, high velocity and agile direction maneuverability. We further extend the capability of bubble control to microrobots for cargo transportation, micropart assembly and transmission of gear structures. We envision this robust bubble manipulation strategy on the PESS would provide a valuable platform for various bubble-involved processes, ranging from microfluidic devices to soft robotics.

Gas bubbles are ubiquitous in natural environments, living organisms, and industrial production, and realizing flexible bubble manipulation in a liquid environment is critical for not only the basic understanding of various processes, ranging from gaseous microreactions to boiling transmission but also a variety of viable applications such as hydrogen production[1–8]. Various techniques leveraging structural and wettability gradients have been put forward to achieve bubble transportation by breaking the symmetric contact line to form a Laplace pressure gradient[9–12]. For instance, inspired by cactus spines and pitcher plants, slippery copper cones with geometry gradients were developed to achieve gas bubble transportation[13–15]. The operating capability mainly originates from the synergistic cooperation of the asymmetric morphology and slippery property of the surface, which help to generate a Laplace force to directionally move the gas bubbles and meanwhile endow a high affinity but low friction force to the bubbles[13]. However, the above techniques are limited by their functional adaptability, such

as short-range bubble transport, fixed bubble moving trajectory, and non-real-time control.

To circumvent these constraints, various external stimuli, including electricity[16–19], magnetism[20–23], and light[24–30] are introduced for bubble maneuvering. Among them, the light-driven bubble manipulation technique has been widely explored for its contactless and high spatiotemporal control. Specifically, non-directional bubble transportation can be achieved by either altering the gas-liquid interface tension or provoking a thermal capillary flow through photothermal effect in a real-time manner[4,24–27,31–33], and these approaches for bubble control mainly resort to bubble asymmetrical deformation[24,27], or Marangoni convection[4,32,33], which always generates a weak driving force for small and slow bubble transportation. This further limits the versatility of efficient bubble detachment and splitting, thereby hindering their application in various practical settings.

[1]State Key Laboratory of High-performance Precision Manufacturing, Dalian University of Technology, Dalian 116024, P. R. China. [2]Shenyang Institute of Automation, Chinese Academy of Sciences, Shenyang 110016, P. R. China. [3]Institutes for Robotics and Intelligent Manufacturing, Chinese Academy of Sciences, Shenyang 110016, P. R. China. ✉e-mail: yahualiu@dlut.edu.cn

In this research, a photopyroelectric slippery surface (PESS) based on the photo-pyroelectric effect is developed to achieve multifunctional bubble manipulation. Due to the generated dielectric wetting and nonuniform electric field under the irradiation of near-infrared (NIR) light, a bubble is subject to both the Laplace force and dielectrophoresis force, enabling its transportation at a high velocity. We demonstrate that bubbles can move precisely along arbitrarily designed paths in an efficient manner with a wide volume range. More importantly, the PESS enables the splitting, merging, and detachment of underwater bubbles, which provides a promising avenue for selective chemical reaction, self-assembly, and cargo transportation.

## Results

### Bubble transport on the PESS

The PESS is constructed with a sandwich structure including a slippery layer, a pyroelectric layer and a photothermal layer from top to bottom (Fig. 1a and Supplementary Fig. 1). Briefly, the mixture of $Fe_3O_4$ nanoparticles and polydimethylsiloxane (PDMS) was streamed on a commercial lithium niobate wafer acting as the pyroelectric layer, and the photothermal layer was obtained by curing the PDMS. Then a coating of hydrophobic $SiO_2$ nanoparticles with a diameter of ~25 nm

(Supplementary Fig. 2a) was sprayed on the other side of the lithium niobate wafer, and the slippery layer was achieved by infusing the silicone oil in the coating. Notably, the clustered $SiO_2$ nanoparticles form a porous structure, which endows a favorable oil storage capacity. Meanwhile, the contact angle is increased with the bubble volume while the sliding angle is always less than 5°, indicating a minor resistance to bubble movement (Supplementary Fig. 2b).

We further examined the photothermal and pyroelectric properties of the PESS, which are closely related to the bubble motion. Figure 1b shows the time-dependent temperature profile on the irradiation point. Specifically, when switching on or off the NIR light, the surface temperature $T$ rises from 20.5 to 69 °C within 5 s or decreases from 85.1 to 44.2 °C within 5 s, leading to a maximum temperature rising and decreasing rate $(dT/dt)_{max}$ of 27.4 °C s⁻¹ and 22.9 °C s⁻¹, respectively. The superior photothermal performance is attributed to the uniform dispersion of $Fe_3O_4$ nanoparticles in the PDMS. Moreover, the intensive and quick temperature variation in the range of 30 °C < $T$ < 90 °C was achieved in more than seven cycles (Supplementary Fig. 3). Taken together, these observations suggest that the PESS offers not only a fast response but also a stable cycle durability for bubble control in potential long-term applications. As an

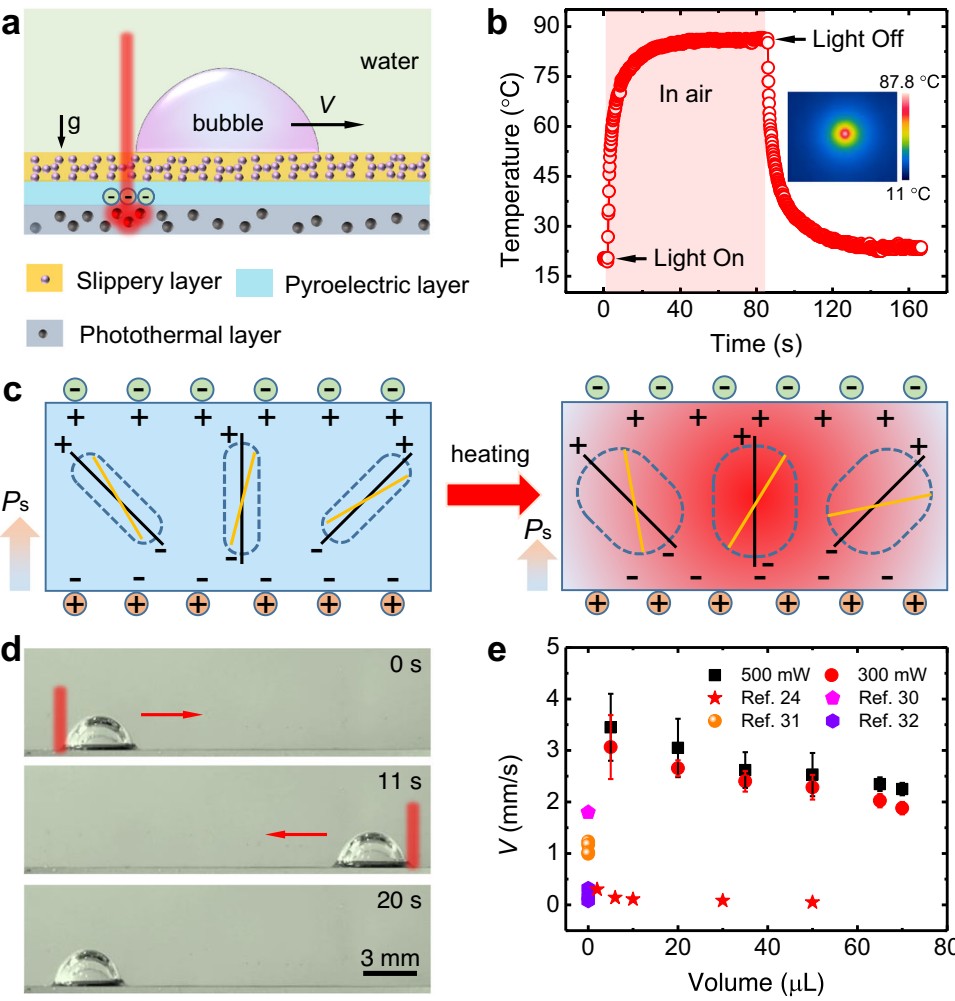

**Fig. 1 | Properties of the PESS and its bubble manipulation. a** Schematic showing the structure of the PESS. As the NIR light irradiates, the photothermal layer produces heat because of the photothermal effect. The temperature within the pyroelectric crystal rises synchronously due to thermal conduction, giving rise to extra surface-free charges, which drive the bubble into motion. **b** Time-dependent temperature profile of the irradiation point on the PESS under 500-mW NIR irradiation. The inset shows the infrared thermal image captured by an infrared camera

at ~80 s. **c** Pyroelectric diagram of the lithium niobate crystal. As the temperature increases, the spontaneous polarization of pyroelectric crystal decreases, giving rise to extra surface-free charges. **d** Reciprocating motion of a bubble on the PESS under the NIR irradiation. **e** Variation of the bubble velocity as a function of volume at different laser power. The error bars of the data denote the standard deviation of three measurements. Source data are provided as a Source Data file.

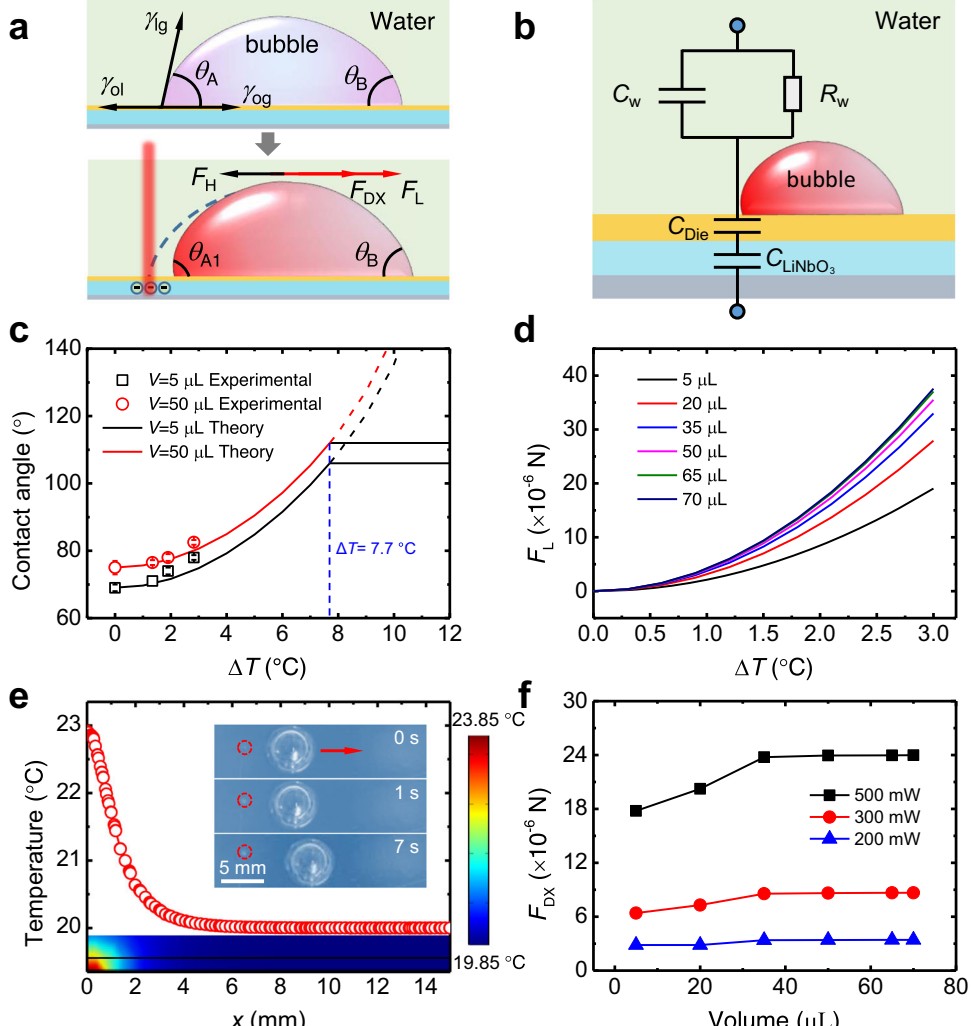

**Fig. 2 | The mechanism of bubble maneuvering on the PESS. a** Schematic showing the force analysis in NIR-induced bubble transport. **b** Schematic showing the equivalent circuit when NIR light irradiates on the left side of the bubble. $C_w$ and $R_w$ are the equivalent capacitance and resistance of water, respectively. **c** Theoretical and experimental values of bubble contact angle as a function of temperature difference at the three-phase contact line. **d** Variation of $F_L$ on bubbles of different volumes with $\Delta T$. **e** Temperature mapping on the PESS was simulated using a finite-element method. The insert shows the image sequences of bubble movement under the irradiation point 3 mm away from the left end of the bubble. **f** $F_{DX}$ as a function of the bubble volume. Source data are provided as a Source Data file.

intermediate layer, the pyroelectric crystal with a pyroelectric coefficient $P_c = 83 \times 10^{-6}$ C m$^{-2}$ K$^{-1}$ rises its temperature simultaneously with that of the photothermal layer when irradiated by an NIR laser due to thermal conduction, leading to a decrease of spontaneous polarization intensity ($P_s$) of the lithium niobate crystal[34,35]. This further lowers the bound surface charges and gives rise to extra surface-free charges (Fig. 1c and Supplementary Fig. 4).

Due to the above synergistic effect of slippery, pyroelectric, and photothermal properties of the PESS, the swift bubble transport on the surface can be realized by controlling the irradiation position of the NIR laser. For example, a 35-μL bubble moves to the right at a velocity of ~2.5 mm s$^{-1}$ when shined on the left side at a power of 500 mW and reverses back when shined on the right (Fig. 1d and Supplementary Movie 1), which is directly opposite to that on a photothermal slippery surface (PSS), where a bubble atop the surface could be driven toward the irradiated spot due to the generated Laplace force by the decrease of the bubble contact angle under NIR irradiation (Supplementary Fig. 5 and Supplementary Discussion 1). The variation of bubble velocity ($V$) versus bubble volume under different laser power indicates that $V$ decreases with the increase of bubble volume and the decrease of laser power (Fig. 1e). Notably, a typical gas bubble of ~5 μL can reach

a velocity over 4.5 mm s$^{-1}$ on the PESS, and 2.28 mm s$^{-1}$ for a large one, e.g., ~50 μL, which is more than an order of magnitude higher compared with that on the PSS[24]. In addition, the anti-buoyancy movement and even movement vertically downwards of bubbles (Supplementary Fig. 6 and Supplementary Movie 2) can be realized due to the strong driving force generated by NIR irradiation. All these results indicate that the PESS possesses superior capability for bubble maneuvering, which needs to be explored further in the light of driving force.

**Bubble driving force analysis on the PESS**

To unravel the mechanism of high-efficiency bubble manipulation on the PESS, we resorted to a force analysis to probe the role of the sandwich structure. Obviously, the symmetrical shape of the bubble on a slippery surface indicates that the left ($\theta_A$) and right ($\theta_B$) contact angles are equal to the apparent contact angle ($\theta$) as $\theta = \theta_A = \theta_B$ (Fig. 2a, Supplementary Fig. 7a and Supplementary Movie 3). With the NIR irradiating on the left side of the bubble, the left side starts to shrink while the right side remains basically unchanged, leading to generated asymmetric contact angles of $\theta_{A1} > \theta_B$. We attributed this phenomenon to the dielectric wetting, which is contrary to the bubble deformation due to the reduction of lubricant surface tension caused by the

temperature rise on the PSS that the bubble contact angle would decrease accordingly[24]. To quantify the acting force during the bubble deformation on the PESS, we consider the traditional dielectric wetting theory, which refers to changing the contact angle between the droplet and the dielectric layer by varying the voltage. According to the Young–Lippmann equation[8], the contact angle with the applied electric field can be described as

$$\cos\theta_1 = \cos\theta_0 + \frac{\varepsilon_0\varepsilon_d}{2\gamma_{LG}d}\Delta U^2 \tag{1}$$

where $\theta_1$ and $\theta_0$ are the water contact angles with and without the applied electric field, respectively, $\varepsilon_0$ and $\varepsilon_d$ are the permittivity of vacuum and the relative dielectric constant of the dielectric layer, respectively, $d$ is the thickness of the dielectric layer, $\gamma_{LG}$ is the surface tension of liquid–gas interface, and $\Delta U$ is the voltage drop across the dielectric layer in the vertical direction at the three-phase contact line (Supplementary Fig. 7b). Note that the zeta potential of bubbles is not taken into account in Eq. (1), due to its negligible effect on dielectric wetting[36–38].

As the NIR light irradiates from the top, it passes through the water layer, the slippery layer, and the pyroelectric layer and readily reaches the underlying photothermal layer. The temperature of the pyroelectric crystal rises rapidly along with the photothermal layer because of thermal conduction (Supplementary Fig. 7c) to decrease the spontaneous polarization intensity of lithium niobate crystal, which further lowers the bound surface charges and gives rise to extra surface free charges, and finally forms the dielectric wetting. Figure 2b shows the equivalent dielectric wetting circuit as the NIR light irradiates on the PESS. Specifically, the pyroelectric crystal and slippery layer can be viewed as two series capacitors that carry an equal amount of charge, the amount of which generated on the surface due to the temperature difference can be expressed by $Q = AP_c\Delta T$, where $A$ and $\Delta T$ are the area of the NIR spot and temperature change, respectively[39]. Considering the voltage drop of the dielectric layer $\Delta U = \frac{Q}{C_{die}} = \frac{P_c\Delta T d}{\varepsilon_0\varepsilon_d}$, where $C_{die} = \frac{A\varepsilon_0\varepsilon_d}{d}$ is the equivalent capacitance of the dielectric layer, the bubble contact angle on the PESS can be expressed by

$$\cos\theta_{A1} = \cos\theta_A - \frac{P_c^2\Delta T^2 d}{2\gamma_{LG}\varepsilon_0\varepsilon_d} \tag{2}$$

which is well confirmed by Fig. 2c and Supplementary Fig. 7d over a wide range of explored laser power and, in turn, supports the rationality of Eq. (2). The asymmetric deformation of the bubble will produce a Laplace force[40] $F_L = 2R\gamma_{LG}(\cos\theta_B - \cos\theta_{A1})$ that drives the bubble away from the light source. In combination with Eq. (2), $F_L$ can be further calculated as

$$F_L = \frac{RdP_c^2\Delta T^2}{\varepsilon_0\varepsilon_d} \tag{3}$$

where $R$ represents the base radius of the bubble on the PESS (Supplementary Fig. 8). According to Eq. (3), the driving force generated by the dielectric wetting is only related to the size of the bubble and the temperature change but independent of bubble contact angles (Fig. 2d).

Note that Eq. (3) only applies when the Young-Lippmann equation is established. Previous research has discussed the saturation phenomenon in dielectric wetting, that is, when the voltage exceeds a certain value, i.e., the saturation voltage, the further increase of voltage will not alter the contact angle. Thus, it is necessary to determine whether the voltage reaches a saturation value during bubble transport. The saturation voltage ($U_S$) can be approximately described by $U_S = [2d\gamma_{OL}/\varepsilon_0\varepsilon_d]^{0.5}$, where $\gamma_{OL}$ is the oil/water surface tension[41]. Here, $U_S$ is estimated to be 144.3 V under a saturation temperature change

$\Delta T = 7.7\,°C$ (Fig. 2c) for $d = 5\,\mu m$ and $\varepsilon_d = 2.5$. Notably, the bubble dynamic response time is about 0.2 s, corresponding to a temperature rise rate of -38.5 °C s$^{-1}$ to reach a saturation voltage. This value is much larger than that in the experiment (Supplementary Fig. 7c), indicating that the saturation voltage would not be reached for current bubble transport on the PESS.

Except for Laplace force, the bubble irradiated by the NIR light may also be subjected to a dielectrophoretic force $F_{DX}$ along $x$ direction due to the generated nonuniform electric field[42] in a nonuniform temperature field (Fig. 2e, Supplementary Fig. 9 and Supplementary Discussion 2). To verify this, we irradiate the point 3 mm away from the left end of the bubble on the PESS, and the irradiated bubble moves far away from the light source immediately (inset in Fig. 2e and Supplementary Movie 4), while the bubble remains stationary on the PSS (Supplementary Fig. 10). Note that, the temperature of the bubble keeps virtually unchanged (Supplementary Fig. 11), indicating that there is no generated $F_L$ according to Eq. (3) because of $\Delta T = 0\,°C$ but $F_{DX}$ that drives the bubble away from the light source. Here, $F_{DX}$ is expressed by the properties of the air inside the bubble and the water surrounding it as

$$F_{DX} = \int\frac{1}{2}\varepsilon_0(\varepsilon_A - \varepsilon_W)\nabla E_X^2 d\Omega \tag{4}$$

where $\varepsilon_A$ and $\varepsilon_W$ are the relative dielectric constant of air and water, respectively, $E_X$ is the electric field intensity along the $x$ direction, and $\Omega$ is the bubble volume[43]. Obviously, the first term $\varepsilon_A - \varepsilon_W < 0$ is expected as the air bubble is submerged in the water, endowing an $F_{DX}$ to propel the bubble to a lower electric field region, i.e., away from the light source.

Figure 2f shows the simulation results of $F_{DX}$ as a function of the bubble volume that $F_{DX}$ increases with increasing laser power and bubble volume, which fits well with the proposed model (Eq. (S10) in Supplementary Discussion 3). Moreover, $F_{DX}$ is estimated to be of the order of 10$^{-6}$ N, which is comparable to that of $F_L$, indicating that the two forces contribute coherently to the bubble transport. In addition, as the bubble moves forward, a backward hydrodynamic resistance force $F_H$ comprising lubricant and liquid media's viscous forces is generated, which could be expressed as[44]

$$F_H \approx \alpha\pi VR(\mu_o + \mu_l) \tag{5}$$

where $\alpha$ is a numerical factor, $\mu_o$ and $\mu_l$ are the viscosity of oil and liquid, respectively. Therefore, the resultant force driving the bubble under NIR irradiation on the PESS is $F = F_L + F_{DX} - F_H$ (Fig. 2a). Obviously, a stationary bubble starts to accelerate after switching on the NIR laser. $F_H$ continues to increase as the bubble speeds up until a balance between the driving force and resistance force, at which point the bubble reaches its steady velocity scaled as $V \sim (F_L + F_{DX})/[R(\mu_o + \mu_l)]$. Moreover, the bubble velocity increases with the laser power but decreases with the bubble volume (Eq. (S20) in Supplementary Discussion 3), which is consistent with the experimental data (Fig. 1e).

## Multifunctional bubble manipulation

The robust bubble maneuvering on the PESS can be harnessed in some unique scenarios, including high-fidelity transport, splitting, and detachment of bubbles, which are not possible with previous strategies. The bubble can move along arbitrary paths with a wide volume range from 0 to 70 μL (Supplementary Fig. 12). Figure 3a shows the stacked images of a 35-μL bubble drawing the trajectory of letter Z (Supplementary Movie 5). Usually, bubble transport in a closed system is more challenging than that in an open one due to the generated larger adhesion force between the upper and lower plates when accelerating a bubble from a static state to a dynamic one[45]. However,

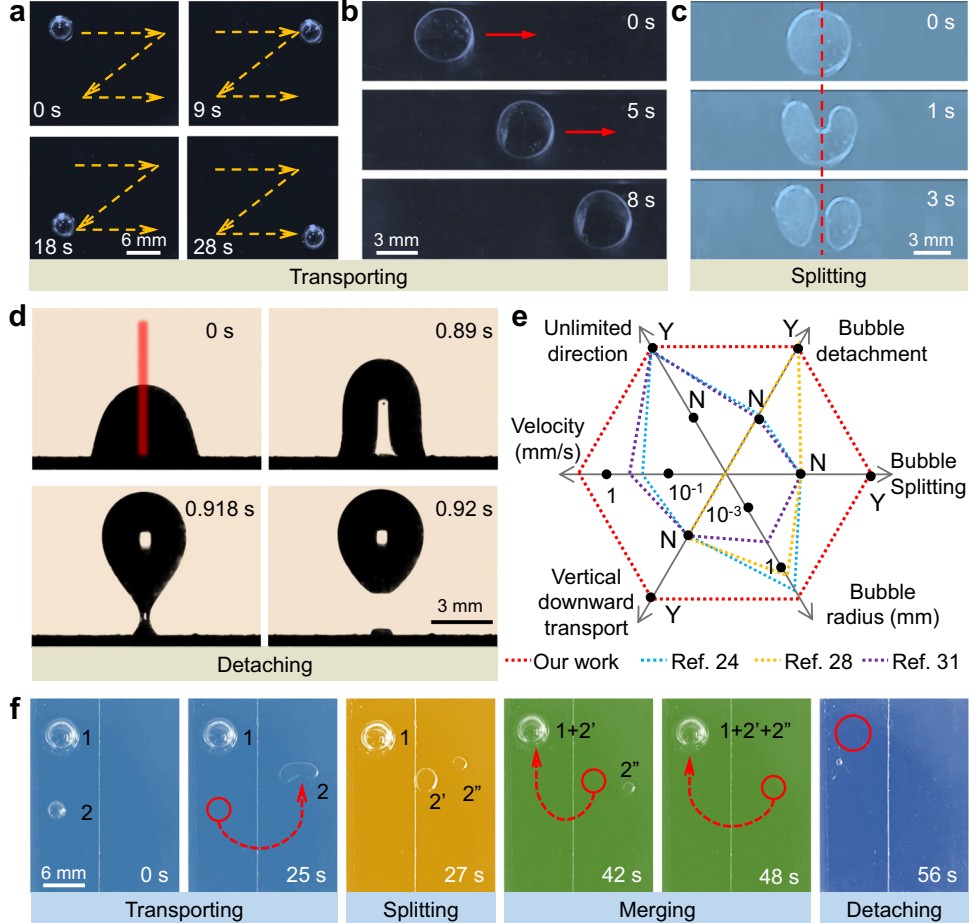

**Fig. 3 | Multifunctional bubble manipulation on the PESS. a** Stacked images of a bubble drawing the trajectory of letter Z. **b** The transportation of bubble compressed by a transparent slippery surface. **c** Sequential images showing the segmentation of a compressed bubble. **d** Sequential images showing the bubble detachment from the PESS. **e** The notable advantages of bubble maneuvering on the PESS compared with other techniques from six aspects. Y, yes; N, no. **f** Versatile bubble manipulation on the PESS, including transporting, splitting, merging, and detaching, successively.

this is achievable on the PESS with a speed of 1.62 mm s$^{-1}$ (Fig. 3b and Supplementary Movie 5), which is in striking contrast to that on the PSS, where the bubble stands still (Supplementary Fig. 13a). Remarkably, further reducing the distance, e.g., smaller than 1 mm, or setting a relatively tiny angle, e.g., smaller than 8°, between the two plates engenders an arbitrary bubble segmentation due to the Laplace pressure towards the interior of the bubble caused by dielectric wetting (Supplementary Discussion 4). Specifically, the bubble depresses inwards after being irradiated by the NIR light, and then the center opens by moving the irradiating spot and splits finally, the phenomenon of which has never been reported before (Fig. 3c, Supplementary Fig. 13b and Supplementary Movie 6). By contrast, the bubble cannot be split on the PSS under the same operating condition due to the Laplace pressure towards the outside of the bubble caused by the photothermal effect (Supplementary Fig. 13c and Supplementary Discussion 4).

Efficient bubble detachment is highly desirable in various practical production, including hydrogen production, heat transfer, and electrochemical processing[46–48]. Figure 3d (Supplementary Fig. 14 and Supplementary Movie 7) shows that a bubble can be readily piloted and forced to quickly detach from the PESS when irradiated at the bubble center, offering a simple and safe approach compared to that with an ultra-high voltage[19,49,50]. It is obvious that the bubble detachment mainly relies on the laser power and bubble volume. In general, a larger bubble necessitates a lower laser power to enable an easier bubble escape and vice versa (Supplementary Fig. 15). This is because

larger bubbles are subjected to a greater upward buoyancy force, correspondingly requiring a smaller additional upward dielectrophoretic force by a lower power laser to overcome the downward capillary force for a bubble detachment (Supplementary Discussion 5). As depicted in Fig. 3e, the PESS has shown superior or even unparalleled performance in six aspects compared with previously reported representative studies using other techniques. "Y" and "N" in the graph indicate yes and no in the literature, respectively. Together, the PESS has shown its universal adaptability for bubble control, which is further demonstrated by a successive in-field bubble manipulation in Fig. 3f. Specifically, bubble 2 is transported to the splitting region, where it is split into two smaller ones, and then they are steered to merge with bubble 1 to form a large one, which eventually detaches from the PESS (Supplementary Movie 8). Note that compared to other bubble manipulation methods[24–30], the mechanism and phenomenon of bubble manipulating on the PESS represents a significant advance in the state of the art on the subject, which is different from the robust droplet control on light-induced charged surfaces[51,52].

## Bubble microrobots
The programmable transport and the arbitrary splitting of bubbles can be further equipped to act as bubble microrobots for cargo transportation, micropart assembly, and transmission of gear structures. For example, a carrier with two droplets could be directed by a bubble to a specified site either through linear motion or rotation by adjusting the irradiation position of the NIR laser (Fig. 4a, b,

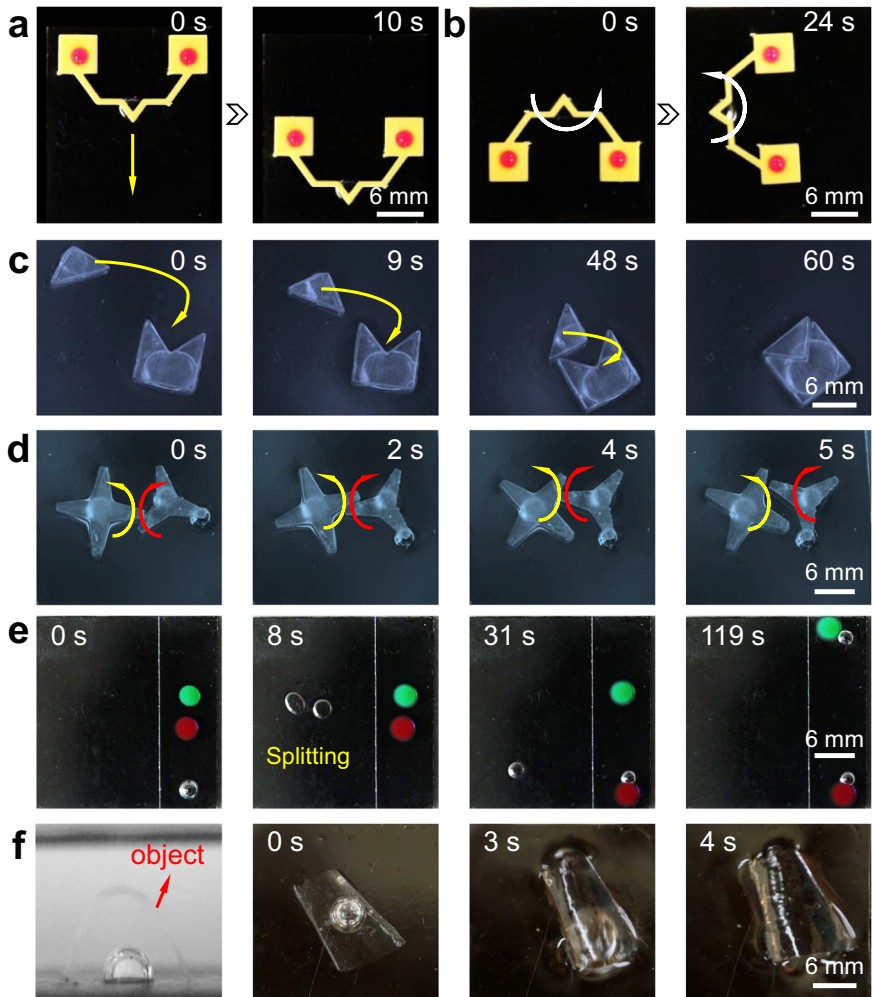

**Fig. 4 | The versatility of bubble microrobots on the PESS. a, b** Image sequences of a bubble microrobot driving the shelves on the water to move in a straight line and rotate. **c** Part assembly at air-water interface realized by bubble microrobots. **d** Gear transmission is driven by bubble microrobots. **e** The cooperative work between small bubble robots formed after the splitting of a large one. **f** Sequential images showing a bubble microrobot to salvage objects in the water.

Supplementary Movie 9). This technique can be further harnessed for assembling matching microparts (Fig. 4c and Supplementary Movie 9) and transmitting paired gears (Fig. 4d and Supplementary Movie 9). Compared to the above functions of bubble microrobots, it is even more magical and practical to realize the splitting of the bubble to do work alone or collaboratively. Figure 4e shows a large bubble microrobot was divided into two tiny ones for sorting hydrogel beads of different colors (Supplementary Movie 9). The bubble robot can also be used for underwater salvage due to its maneuverable capability. After a bubble detached from the PESS, it would adhere to the target object and move to the water surface together under the bubble buoyancy (Fig. 4f, Supplementary Movie 9). These pioneering developments offer exciting opportunities for bubble-based robotics.

## Discussion

In summary, a PESS has been proposed for versatile bubble maneuvering by leveraging both the Laplace force and dielectrophoresis force. On the designed surface, the bubble can move along arbitrary paths in an efficient manner with a wide volume range under NIR irradiation. Benefiting from the generated strong driving force, the PESS grants the bubble remarkable abilities, including splitting, merging, and high-efficiency detachment. Our work further demonstrates that the high-fidelity bubble control on the PESS can be harnessed as bubble microrobots for cargo transportation, micropart assembly, and transmission of gear structures, which would open promising avenues

for bubble manipulation in diverse applications, ranging from microfluidic systems to heat transfer and microoperation.

## Methods

### Materials and preparation of the PESS

$Fe_3O_4$ nanoparticles with a diameter of ~10 nm were obtained from Tianjin Kaili Metallurgical Research Institute (Tianjin, China). PDMS silicone elastomer was purchased from Dow Corning (Sylgard 184, USA), which is used as the host matrix. Silicone oil (20 and 100 mPa s) and ethanol were purchased from Meryer Chemical Technology Co. Ltd. (Shanghai, China). Z-cut optical-grade lithium niobate wafers with a thickness of 0.5 mm were provided by Rayon Optical Materials (Shanghai, China). A commercial solution of superhydrophobic $SiO_2$ nanoparticles was purchased from Rust-Oleum (USA). Quartz glass sheets were purchased from Beijing Zhong Cheng Quartz Glass Co. Ltd (Beijing, China). Deionized water (resistivity ~18 MΩ, 1 mPas) was derived from a Mili-Q water purification system (Summer-S2-20H, Sichuan Delishi Technology Co., Ltd., China). The PESS is constructed with a sandwich structure, including a slippery layer, a pyroelectric layer, and a photothermal layer from top to bottom.

### Preparation of the photothermal layer

$Fe_3O_4$ nanoparticles, PDMS prepolymer, and PDMS curing agent at a mass ratio of 0.6:1:0.1 were mechanically mixed for 10 min at 1000 rpm. Then the prepared mixture was placed in a vacuum

chamber for 20 min to remove air bubbles, followed by spin-coated on the lithium niobate crystal wafers (Supplementary Fig. 1) or quartz glass sheets for 60 s at 1000 rpm. Last, the samples were cured at 80 °C for 2 h.

## Preparation of the slippery layer

Hydrophobic $SiO_2$ nanoparticles were uniformly sprayed on the other side of lithium niobate crystal wafers or quartz glass sheets to form a superhydrophobic layer (Supplementary Fig. 1). Then, the silicone oil was spin-coated on the superhydrophobic layer for 20 s at 500 rpm to form the slippery layer. As a result, the PESS or PSS were fabricated, respectively.

## Bubble manipulation on the PESS/PSS

Two methods were proposed for introducing bubbles into the bubble manipulation system. One is to directly inject bubbles using an injector, where the bubble size can be accurately controlled. The second one is generating surface bubbles in water by leveraging the photothermal effect[33] (Supplementary Fig. 16), which is only used for underwater salvage in the experiment. The near-infrared laser with a spot size of 1 × 1 mm is provided by Fuzhe Technology Co., Ltd., China. Specifically, 808-nm NIR lasers in a Gaussian beam profile with different powers of 100, 200, 300, 500, and 1000 mW were used for the bubble manipulation, corresponding to the laser models of FU808AD100-16GD, FU808AD200-16GD, FU808AD300-16GD, FU808AD500-16GD, and FU808AD1000-16GD, respectively. The irradiation distance between the laser tip and the surface was fixed at 10 cm in the experiment. The trajectory of bubbles can be controlled in a real-time manner by changing the position of the NIR light. The bubble manipulation process was captured using a single-lens reflex camera (EOS 5D MarkIV, Cannon, Japan) or a high-speed camera (Fastcam SA4, Photron, Japan).

## Surface characterization

The surface morphology of the surface before being perfused with silicone oil was characterized by using a field-emission scanning electron microscope (SEM, JSM-6700F, Japan). The contact angle and sliding angle of the bubble on the surface were measured using an OCA25 Standard Contact Angle Goniometer (Dataphysics GmbH, Filderstadt, Germany). At least three measurements were performed on each surface. The temperature of the surface was monitored by a thermal infrared camera (280, Fotric, China) and thermoelectric coupling thermometer (JK3016, Changzhou JinKo Electronic Technology Co. Ltd., China).

## Data availability

The data that support the findings of this study are available from the corresponding authors upon request. Source data are provided in this paper.

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

## Acknowledgements

This work was supported by the National Key Research and Development Program of China (2022YFB4602401, Yahua Liu), the National Natural Science Foundation of China (52075071, Yahua Liu), the Opening Project of the Key Laboratory of Bionic Engineering (Ministry of Education), the Jilin University (KF20200002, Yahua Liu), and the Key Laboratory of Icing and Anti/De-icing of CARDC (IADL20210405, Yahua Liu).

## Author contributions

Y. Liu and H.Z. conceived the research. Y. Liu supervised the research. H.Z., Z.Y., Y. Li, and L. Zhang carried out the experiment. H.Z., H.L., Y.Z., and Z.W. performed the characterization. Y. Liu, S.F., L. Zhao, and H.Z. built the models. All authors analyzed the data and wrote the paper.

## Competing interests

The authors declare no competing interests.
