## [Peer Review File · Nature Communications]

REVIEWER COMMENTS

Reviewer #1 (Remarks to the Author):

The authors present an experimental and theoretical/numerical study on the manipulation of a single bubble sitting on a photopyroelectric slippery surface by means of a NI laser beam. The authors demonstrate how a combination of dielectrophoretic force and Laplace force can be used to promote the movement of a bubble (relative to the individual application of those forces) but also to alter the bubble morphology.

In general, I find the topic very interesting from a fundamental point of view and potentially useful for some industrial and medical applications. The manuscript reads well and the information in the figures are nicely presented. The ideas behind the phenomenon are illustrated with explanatory schemes which are properly combined with high quality experimental images. The theoretical formulations are well described and the experimental results are properly justified.

The idea of moving bubbles attached to small objects with optical tweezers or electrowetting, as to produce a functionalization of the bubbles by attaching them to small objects or particles has been around for at least two decades. In the present study, the novelty factor is thus given by the use of a PESS substrate in combination with the laser to further accelerate the bubble displacement respect to the obtained in slippery lubricant-infused porous surface (SLIPS) driven only by a photothermal effect. However, this technique has been used in a very similar way to manipulate, and merge drops. Some clear examples containing very similar experiments and verification trials using a drop instead of a bubble can be found in:

- Fang Wang et al., Light-induced charged slippery surfaces. *Sci. Adv.* 8, eabp9369 (2022)

- Wang, F. et al. Light control of droplets on photo-induced charged surfaces. *Natl. Sci. Rev.* 10, nwac164 (2023).

These works should be cited in the main text of the draft. At the same time, the many similarities found between those and the current manuscript should be stressed, as their visibility is rather poor in the supplemental material. In the light of these previous reports, I think that the work would greatly benefit for further discussion on the limitations of the method and the level of improvement relative to the state of the art. The authors should make clear why the current work is not a variant of a known method, which has similar working principles (as the one presented in this new study) but it is now applied to a different media. In my opinion, the authors need to account for differences on the physical mechanics behind both cases, i.e. bubble manipulation and drop manipulation with laser on a PESS, and how they represent a significant advance in the state of the art of the topic.

Some parts of the manuscript, specially the second half, reports mostly qualitative results that could be treated in a more rigorous or quantitative way. In particular, the authors should not limit to report some experimental observations but to give at least some rough idea on the physics behind them.

The methods section leaves out some critical details on the PESS sample preparation and the equipment characteristics. This information is fundamental to guarantee reproducibility of the experimental technique.

This is an interesting topic and the results included in the manuscript are surely worth being shared with and deserves publication. However, I can not recommend it for publication in Nature Communications in its current state. The previous remarks and the ones listed below need to be clarified.

Some specific questions and remarks:

1-In some parts of the manuscript the supplemental material is used as an integral part of the work. Thus, entire sentences of the main text are not possible to follow without looking at the supplemental material. Considering that the main text should be "stand alone" I recommend to try including the referenced figures in the main text or remove the explicit dependencies, to make the paper readable yet without looking at the supplemental text.

2-The correlation between the bubble speed V and its volume is properly explained by the difference in net force computed for each case, but what is the origin of that difference? the relative size of the area illuminated by the laser respect to the bubble size? Is the system scalable? I guess that the surface tension would break the scalability but it would be nice if the authors discuss the limitations of the method in this regard.

3-Following the previous question. The authors should clarify why the transport in a closer system is harder than in an open one. Is it due to the viscous friction with the upper plate? How does the friction scales with the bubble size? Would be possible to use semi-transparent PESS to induce a force in both sides of the bubble? (i.e. top and bottom). Closely related, the mechanisms of bubble splitting needs also further explanation.

4-It would be interesting that the authors further discuss the origin of the discrepancy between the expected saturation voltage (V_s) estimated by and the one verified by the experiments. This also applies to the saturation temperature. As the whole phenomenon is thermally driven I think that this might be a relevant issue to explain.

5-It would be helpful for the reader to include the direction of the gravitation acceleration in Figure 1 (a) to make clear that the bubble is sitting on the PESS.

6-In the caption of Figure 1 (b), the authors could clarify that the thermal image is a picture and not a COMSOL simulation.

7-Are there any permanent changes as a consequence of the repeated localized heating and cooling cycles applied to the PESS? e.g. silicon evaporation or porous structure cracks produced by thermal stresses.

8-A bubble velocity of 4.5 mm s^{-1} described as “ultrafast” is questionable. Previous works with drops reports more than 33 mm s^{-1}

9-“In general, a larger bubble requires a lower laser power to enable an easier bubble escape and vice versa.” This should be explained not only on the Supplemental material.

10-In the section dedicated to “microrobot” the authors state all the benefits of the technique but they omitted to mention how the bubbles could be introduced in the system on the first place.

11-What is the beam profile of the laser? What is the laser power and precise wavelength? Model??

Reviewer #2 (Remarks to the Author):

Report on the paper by Zhan et al.

It is an excellent paper on versatile bubble maneuvering on photopyroelectric slippery surfaces. However, there are some unclear points.

1. Line 32-34, “For instance, inspired by cactus spines and pitcher plants, slippery copper cones with geometry gradient were developed to achieve the gas bubble transportation.” It is required to cite some references for this statement.

2. With regard to the manipulation on a photothermal slippery surface (PSS) shown in Supplementary Fig. 5, it is required to explain the mechanism of the manipulation in the manuscript.

3. It is required to explain in the manuscript why zeta potential of a bubble is not taken into account in eq. (1).

4. With regard to eq. (S10), dimension of $F R^{-1}$ is kg s^{-2} which differs from the dimension of velocity m s^{-1} . Thus eq. (S10) seems to be wrong.

Point-to-point response to reviewers' comments

Reviewer #1 (Remarks to the Author):

The authors present an experimental and theoretical/numerical study on the manipulation of a single bubble sitting on a photopyroelectric slippery surface by means of a NI laser beam. The authors demonstrate how a combination of dielectrophoretic force and Laplace force can be used to promote the movement of a bubble (relative to the individual application of those forces) but also to alter the bubble morphology.

In general, I find the topic very interesting from a fundamental point of view and potentially useful for some industrial and medical applications. The manuscript reads well and the information in the figures are nicely presented. The ideas behind the phenomenon are illustrated with explanatory schemes which are properly combined with high quality experimental images. The theoretical formulations are well described and the experimental results are properly justified.

Response: We really appreciate the reviewer's positive feedback and the time she/he has taken to make constructive comments on our work.

The idea of moving bubbles attached to small objects with optical tweezers or electrowetting, as to produce a functionalization of the bubbles by attaching them to small objects or particles has been around for at least two decades. In the present study, the novelty factor is thus given by the use of a PESS substrate in combination with the laser to further accelerate the bubble displacement respect to the obtained in slippery lubricant-infused porous surface (SLIPS) driven only by a photothermal effect. However, this technique has been used in a very similar way to manipulate, and merge drops. Some clear examples containing very similar experiments and verification trials using a drop instead of a bubble can be found in:

- Fang Wang et al., Light-induced charged slippery surfaces. *Sci. Adv.* 8, eabp9369 (2022)

- Wang, F. et al. Light control of droplets on photo-induced charged surfaces. *Natl. Sci. Rev.* 10, nwac164 (2023).

These works should be cited in the main text of the draft. At the same time, the many similarities found between those and the current manuscript should be stressed, as their visibility is rather poor in the supplemental material.

Response: We completely agree with the reviewer that these two pertinent research should be cited in the main text but not only in the Supplementary Information. Following the reviewer's suggestion, we have added some words in the revised manuscript to highlight the significance of these two researches as highlighted in red "...Note that compared to other bubble manipulation methods²⁴⁻³⁰, the mechanism and phenomenon of bubble manipulating on the PESS represents a significant advance in the state of the art on the subject, which are different from the robust droplet control on light-induced charged surfaces^{51, 52}."

In the light of these previous reports, I think that the work would greatly benefit for further discussion on the limitations of the method and the level of improvement relative to the state of the art.

Response: We thank the reviewer for this valuable comment. The main point of this research lies in providing a new design paradigm for versatile bubble maneuvering in various practical applications. Distinct from previous research, in this work, a photopyroelectric slippery surface (PESS) is constructed with a sandwich structure including a slippery layer, a pyroelectric layer and a photothermal layer from top to bottom. Due to the generated dielectric wetting and nonuniform electric field under the near infrared (NIR) irradiation, a bubble is subject to both the Laplace force and dielectrophoresis force, enabling a high-efficiency bubble steering. We demonstrate that the splitting, merging and detachment of underwater bubbles can be achieved with high flexibility and precision, high velocity and agile direction maneuverability. We further extend the capability of bubble control to microrobots for cargo transportation, micropart assembly and transmission of gear structures. We believe that the mechanism of versatile bubble manipulation on the photopyroelectric slippery surface provides a new dimension to our ability to control bubble dynamics and advances the process of intelligent slippery surfaces from simplification of functions to diversification. This solves the problem of single bubble control function, and develops bubble robots, etc., which will bring breakthroughs in fields such as microfluidic chips.

Notably, the photopyroelectric slippery surface is constructed with a sandwich structure with a lubricant layer at the top surface. Due to the inevitable loss of lubricant layer caused by evaporation, bubble displacement and shear flow, the as-fabricated surfaces maybe not qualified enough for sustainable bubble manipulation. This challenge was partially addressed by designing an intelligent slippery surface in our recent publication, i.e., *Adv. Funct. Mater.* 2023, **33**, 2211317. The durability of the surface is rendered by the synergistic effect of the interconnected porous structure and the photothermal expansion of the material. When the lubricant on surface is lost, the photothermal expansion of materials can quickly squeeze the lubricant inside the base to flow out of the interconnected porous structure to generate a fresh lubricant layer to support the sustainable mobility of droplets. Attractively, when the NIR light is turned off, the rebuilt lubricant layer can be swiftly self-absorbed into the porous to inhibit unnecessary wastage.

Furthermore, the property of the surrounding liquid damages the scalability of the system, as discussed in our response to Comment 2 of the reviewer. For example, low-surface-tension surrounding liquid would reduce the confinement of surface bubbles in the vertical direction or create a lubrication failure of the PESS, resulting in a bubble manipulation in a reduced volume range. In addition, this technique is limited in the implementation of bubble segmentation operation and small bubble detachment difficulties. For example, two plates are required to constrain the bubble motion with proper distance or angle between them for bubble splitting. However, considering the main focus of our work is the versatile bubble maneuvering leveraging both the Laplace force and dielectrophoresis force rendered by the photopyroelectric slippery surfaces, we would like to explore these points in our future study, which is related to various practical applications.

The authors should make clear why the current work is not a variant of a known method, which has similar working principles (as the one presented in this new study) but it is now applied to a different media. In my opinion, the authors need to account for differences on the physical mechanics behind both cases, i.e. bubble manipulation and drop manipulation with laser on a PESS, and how they represent a significant advance in the state of the art of the topic.

Response: As pointed out by the reviewer, the concept of using light-thermal-electric field conversion for droplet manipulation has been previously reported^{1,2}. Additionally, this approach finds extensive application in manipulating micro-nano particles^{3,4} in microfluidic systems. Here, for the first time, we show that this method can be applied for versatile bubble manipulation, which extends its application scope, and the underlying mechanism for high-efficiency bubble steering is unveiled leveraging the synergism of two driving forces. More importantly, we demonstrate that this method can be harnessed in some unique scenarios which were previously inaccessible with existing bubble manipulation methods. In this regard, our work is not a following up or incremental work, but represents a significant advance in the state of the art of the topic, which yields a superior platform for multifunctional bubble manipulation in various practical settings. In the following, we will briefly discuss the difference of this method applied for droplet and bubble manipulation in terms of mechanism, phenomena, and application.

Difference in mechanism. As discussed in previous research¹, an irradiated droplet on the photopyroelectric slippery surface is subject to two forces, i.e., a Marangoni force F_M , due to the temperature gradients to drive the droplet towards a low temperature field, and a dielectrophoretic force F_E resulting from the non-uniform electric field. Note that, the direction of the dielectrophoretic force depends on the position of the droplet relative to the irradiation point, which can be directed towards or away from the light source by controlling the irradiation position, as shown in Fig. R1. Therefore, these two forces act in a synergistic or competitive manner to control the droplet moving direction, either following or moving away from the light source. However, an irradiated bubble on the photopyroelectric slippery surface is driven away from the light source by leveraging a Laplace force and a dielectrophoretic force. Specifically, the former one is originated from the asymmetric bubble deformation because of dielectric wetting to drive the bubble far away from the light source. The latter one is generated by the non-uniform electric field. In comparison to the above position-dependent direction of dielectrophoretic force for droplet control, this force for bubble maneuvering is always directed to away from the light source. Therefore, a high-efficiency bubble steering far away from the light source is achieved through the synergistic cooperation of the Laplace force and the dielectrophoretic force. In summary, droplet and bubble transport on photopyroelectric slippery surface are different in terms

of both the driving forces and their corresponding contributions.

Fig. R1 | Mechanism of droplet dynamics on the light-induced charged slippery surfaces¹. **a** $F_E = F_M = 0$ is attained when the NIR light irradiates far away from the left side of the droplet ($L > R_d + R_0$). **b** When the NIR light irradiates the left side of the droplet ($L \approx R_d + R_0$), the electrophoretic force F_E directs towards the high temperature region, which is opposite to the direction of Marangoni force F_M . **c** When the left edge of droplet is close to the laser spot center ($0 < L \leq R_d$, or $L \approx R_d - R_0$), F_E points towards the low temperature region, which is in the same direction of F_M . **d** $F_E = F_M = 0$ is attained when the droplet center coincides with the laser spot center ($L \approx 0$). Here, R_0 , R_d and L_0 are the width of laser spot, the droplet radius and the distance between the laser spot center and the droplet center, respectively.

Difference in phenomena. Different phenomena can be observed when a droplet and a bubble are irradiated by the NIR laser on the photopyroelectric slippery surface. Specifically, a droplet can be attracted to or repulsed away from the light source by varying the distance between the light source and the droplet. However, a bubble always moves away from the light source by leveraging the synergistic cooperation of the Laplace force and dielectrophoresis force. Moreover, fast bubble detachment from the surface and splitting into smaller ones were achieved on the surface, which show extraordinary potential in various applications. In addition, compared to droplet manipulation requiring the introduction of droplets from the outside, the bubbles can

be generated in-situ from within the system through photothermal effects, which reduces the disturbance to the system.

Difference in applications. Due to the different dynamic behaviors of droplets and bubbles under NIR irradiation, they can be used in different application domains. For example, droplet manipulation on photoelectric slippery surfaces is more prevalent in the fields such as microfluidic, drug delivery and inkjet printing, while the versatility of bubble manipulation allows it to be applied in wastewater treatment, energy utilization, and boiling heat transfer.

In addition to the difference compared to droplet manipulation, bubble maneuvering on photopyroelectric slippery surfaces suggests a new design paradigm in comparison with existing bubble manipulation techniques. **New phenomena:** Compared with other strategies that only allow for single bubble manipulation, the PESS can achieve fast programmable transport, splitting and detachment of bubbles under the NIR irradiation, which holds tremendous significance in practical settings. **New mechanism:** Unlike previous reports that light-controlled bubbles rely on single weak Marangoni effect or asymmetrical deformation, the robust bubble manipulation on the PESS is underpinned by the Laplace force and dielectrophoresis force generated by dielectric wetting and nonuniform electric field under NIR irradiation, which provides a new dimension to our ability to control bubble dynamics. **New applications:** The realization of multifunctional bubble manipulation on the PESS shows extraordinary potential in various applications including microfluidic devices and microrobots.

In this regard, it is fair to say that our work is not a variant of a known method, but represents a considerable advance in bubble manipulation.

Some parts of the manuscript, specially the second half, reports mostly qualitative results that could be treated in a more rigorous or quantitative way. In particular, the authors should not limit to report some experimental observations but to give at least some rough idea on the physics behind them.

Response: Following the reviewer's suggestion, we have added more detailed analysis to explain the experimental observations, including the mechanism of bubble manipulation on photothermal slippery surfaces (PSS), and the mechanism of bubble

splitting on photopyroelectric slippery surfaces.

• **Mechanism of bubble manipulation on photothermal slippery surfaces**

As reported previously⁵, the bubble transportation on photothermal slippery surfaces under NIR irradiation is attributed to the Laplace force generated by the NIR-induced asymmetric deformation of bubbles. Initially, the left and right contact angles of the bubble on the PSS are equal to each other without irradiation as $\theta_{a0}=\theta_{b0}=\theta$ (Fig. R2), which can be expressed by

$$\cos \theta = \frac{\gamma_{OL}-\gamma_{OG}}{\gamma_{LG}} \tag{R1}$$

where γ_{OG} , γ_{OL} and γ_{LG} are the tensions of the oil-gas, oil-liquid and liquid-gas interfaces, respectively. When the NIR light irradiating on the left side of the bubble, the temperature of the illuminated position rises quickly, which leads to a decreased γ_{OG} , corresponding to a decreased θ_{a0} while θ_{b0} remains relatively unchanged (Fig. R2) and a asymmetric deformation of the bubble. This asymmetric deformation of the bubble generates a Laplace force $F_L=2R\gamma_{LG}(\cos \theta_{a1} - \cos \theta_{b0})$ that drives the bubble to chase the light source. By contrast, the surface tension change can be neglected on the PESS due to the low temperature rise. We have added this information in the revised manuscript as highlighted in red: “which is directly opposite to that on a photothermal slippery surface (PSS), where a bubble atop the surface could be driven toward the irradiated spot **due to the generated Laplace force by the decrease of the bubble contact angle under NIR irradiation** (Supplementary Fig. 5 and Supplementary Discussion I).”.

Fig. R2 | Schematics showing the force analysis in NIR-induced bubble transport on the photothermal slippery surface. F_H is the resistance force.

• **Mechanism of bubble splitting on photopyroelectric slippery surfaces**

(1) Bubble splitting between two parallel plates. As shown in Fig. R3a, when compressed by a parallel plate, a bubble on the PESS forms a concave surface, which is subjected to a uniform outward Laplace force in the radial direction. Note that, the water contact angles at the upper (θ_{u0}) and lower (θ_0) surface are equal to each other, i.e., $\theta_{u0} = \theta_0$. When the NIR light irradiating left side of the bubble, as shown in Fig. R3b, the water contact angle θ on the bottom surface decreases due to the generated dielectric wetting⁶, which could be expressed as

$$\cos \theta_1 = \cos \theta_{10} + \frac{\varepsilon_0 \varepsilon_d}{2\gamma_{LG}d} \Delta V^2 = \cos \theta_{10} + \frac{P_c^2 \Delta T^2 d}{2\gamma_{LG} \varepsilon_0 \varepsilon_d} \quad (\text{R2})$$

where θ and θ_0 are the contact angles of water and the lower surface with and without NIR irradiation, respectively, ε_0 and ε_d are the permittivity of vacuum and the relative dielectric constant of the dielectric layer, respectively, d is the thickness of the dielectric layer, and $\Delta V = \frac{P_c \Delta T d}{\varepsilon_0 \varepsilon_d}$ is the voltage drop across the dielectric layer in the vertical direction at the three-phase contact line with P_c being pyroelectric coefficient and ΔT the temperature change.

Fig. R3 | Experimental image and schematics showing the mechanism of bubble splitting between two parallel plates without (a) and with (b) NIR irradiation.

Obviously, after switching on the NIR light, a Laplace pressure towards the inside of the bubble generates at the left side due to the transformation of the concave meniscus (Fig. R3a) to a convex one (Fig. R3b) to split the bubble, which can be expressed as

$$P_0 = \gamma_{LG} \left(\frac{1}{r_1} + \frac{1}{r_2} \right) \quad (\text{R3})$$

where P_0 is the pressure of the bubble, r_1 and r_2 are the principal radii of curvature, and

r_2 is equal to the width of laser spot $R_0 = 0.5$ mm. The radius of the meniscus of the bubble r_1 is geometrically related to the local contact angles and the distance between the plates d_h , which could be expressed as

$$r_1 = \frac{d_h}{\cos \theta_u + \cos \theta_l} \quad (\text{R4})$$

where θ_u is the contact angle of the water with the upper surface after NIR irradiation. A combination of Equations (R3) and (R4) leads to the Laplace pressure to split the bubble as

$$P_0 = \gamma_{LG} \left(\frac{\cos \theta_u + \cos \theta_l}{d_h} + \frac{1}{R_0} \right) \quad (\text{R5})$$

It is clear from Equation (R5) that the Laplace pressure for bubble splitting increased after reducing the distance between two plates d_h , which is in line with the experimental results. Notably, the dielectrophoretic force shows little effect on the bubble splitting because it acts on the whole bubble^{7,8}.

Fig. R4 | Schematic showing bubble splitting process on the PESS without (a) and with (b) NIR irradiation.

(2) **Bubble splitting between two inclined plates.** When a bubble is compressed by an inclined plate on the PESS, a Laplace force is generated to drive the bubble to the tip⁹ (Fig. R4a), and under the NIR irradiation, the bubble deforms to generate a Laplace pressure towards the inside of the bubble to splitting the bubble (Fig. R4b), which can be calculated by

$$P_0 = \gamma_{LG} \left(\frac{\cos \theta_u + \cos \theta_l}{D_1 \tan \beta} + \frac{1}{R_0} \right) \quad (\text{R6})$$

where D_1 is the bubble length in the horizontal direction, β is the angle between two plates. It is clear that the Laplace pressure for bubble splitting increases at a reduced β , which is in line with the experimental result. We have added this information in the revised manuscript as highlighted in red: "...engenders an arbitrary bubble

segmentation due to the Laplace pressure towards the interior of the bubble caused by dielectric wetting (Supplementary Discussion IV).”.

By contrast, the contact angle of a bubble compressed by a horizontal plate or an inclined plate on the PSS under NIR irradiation becomes smaller⁵ (Fig. R5), corresponding to a larger Laplace pressure $P_0 = \gamma_{LG} \left(\frac{\cos \theta_u + \cos \theta_l}{-d_h} + \frac{1}{R_0} \right)$ or $P_0 = \gamma_{LG} \left(\frac{\cos \theta_u + \cos \theta_l}{-D_1 \tan \beta} + \frac{1}{R_0} \right)$, respectively, towards the outside of the bubble to further extend the bubble, during which the bubble splitting is impossible. We have added this information in the revised manuscript as highlighted in red: “By contrast, the bubble cannot be split on the PSS under the same operating condition, due to the Laplace pressure towards the outside of the bubble caused under the photothermal effect (Supplementary Fig. 13c and Supplementary Discussion IV).”.

Fig. R5 | Schematic showing the force of a compressed bubble on the PSS under NIR irradiation. **a** Bubble compressed by a horizontal plate on the PSS. **b** Bubble compressed by an inclined plate on the PSS.

The methods section leaves out some critical details on the PESS sample preparation and the equipment characteristics. This information is fundamental to guarantee reproducibility of the experimental technique.

Response: We thank the reviewer for pointing out this issue. Following the reviewer’s suggestion, we have supplemented more details on the PESS preparation and the

equipment characteristics in the Methods part of the revised manuscript as

“Preparation of the PESS. The photopyroelectric slippery surface is constructed with a sandwich structure including a slippery layer, a pyroelectric layer and a photothermal layer from top to bottom.

Preparation of the photothermal layer. Fe₃O₄ nanoparticles, PDMS prepolymer and PDMS curing agent at a mass ratio of 0.6:1:0.1 were mechanically mixed for 10 min at 1000 rpm. Then, the prepared mixture was placed in a vacuum chamber for 20 min to remove air bubbles, followed by spin-coated on the LiNbO₃ crystal wafers (Supplementary Fig. 1) or quartz glass sheets for 60 s at 1000 rpm. Last, the samples were cured at 80 °C for 2 hours.

Preparation of the slippery layer. Hydrophobic SiO₂ nanoparticles were uniformly sprayed on the other side of LiNbO₃ crystal wafers or quartz glass sheets to form a superhydrophobic layer (Supplementary Fig. 1). Then, the silicone oil was spin-coated on the superhydrophobic layer for 20 s at 500 rpm to form the slippery layer. As a result, the photopyroelectric slippery surface (PESS) or photothermal slippery surface (PSS) were fabricated, respectively.

Bubble manipulation on the PESS/PSS. Two methods were proposed for introducing bubbles into the bubble manipulation system. One is to directly inject bubbles using an injector, where the bubble size can be accurately controlled. The second one is generating surface bubbles in water by leveraging the photothermal effect³³ (Supplementary Fig. 16), which is only used for underwater salvage in the experiment. The near infrared laser with a spot size 1×1 mm is provided by Shenzhen Fuzhe Technology Co., Ltd., China. Specifically, 808-nm NIR lasers in a Gaussian beam profile with different powers of 100 mw, 200 mw, 300 mw, 500 mw, and 1000 mw were used for the bubble manipulation, corresponding to the laser models of FU808AD100-16GD, FU808AD200-16GD, FU808AD300-16GD, FU808AD500-16GD, and FU808AD1000-16GD, respectively. The irradiation distance between the laser tip and the surface was fixed at 10 cm in the experiment. The trajectory of bubbles can be

controlled in a real-time manner by changing the position of the NIR light. The bubble manipulation process was captured using a single-lens reflex camera (EOS 5D MarkIV, Cannon, Japan) or a high-speed camera (Fastcam SA4, Photron, Japan).

Characterizations. The surface morphology of the surface before being perfused with silicone oil was characterized by using a field-emission scanning electron microscope (SEM, JSM-6700F, Japan). The contact angle and sliding angle of the bubble on the surface were measured using an OCA25 Standard Contact Angle Goniometer (Dataphysics GmbH, Filderstadt, Germany). At least five measurements were performed on each surface. The temperature of the surface was monitored by a thermal infrared camera (280, Fotric, China) and thermoelectric coupling thermometer (JK3016, Changzhou JinKo Electronic Technology Co. LTD, China).”.

This is an interesting topic and the results included in the manuscript are surely worth being shared with and deserves publication. However, I can not recommend it for publication in Nature Communications in its current state. The previous remarks and the ones listed below need to be clarified.

Response: Again, we thank the reviewer for her/his appreciation of our work. In the following, we respond point-by-point to each of the specific questions and remarks.

Some specific questions and remarks:

1-In some parts of the manuscript the supplemental material is used as an integral part of the work. Thus, entire sentences of the main text are not possible to follow without looking at the supplemental material. Considering that the main text should be “stand alone” I recommend to try including the referenced figures in the main text or remove the explicit dependencies, to make the paper readable yet without looking at the supplemental text.

Response: We thank the referee for her/his comment on improving the presentation of our writing. In line with the reviewer’s comment, we have carefully rephrased the manuscript to improve the readability of the revised manuscript. Moreover, the manuscript has been carefully edited to maximize the accessibility and therefore the

impact of our work.

2-The correlation between the bubble speed V and its volume is properly explained by the difference in net force computed for each case, but what is the origin of that difference? the relative size of the area illuminated by the laser respect to the bubble size? Is the system scalable? I guess that the surface tension would break the scalability but it would be nice if the authors discuss the limitations of the method in this regard.

Response: We thank the reviewer for this insightful comment. There are several points embedded in this comment, for which we would like to address one by one.

(1) The high-efficiency bubble manipulation is achieved by leveraging both the Laplace force and dielectrophoresis force, resulting from the dielectric wetting and non-uniform electric field, respectively. Note that, both the dielectric wetting and non-uniform electric field originated from the temperature rise of the contact surface between the bubble and the PESS under the NIR irradiation. As we discussed in the Supplementary Materials of the manuscript, the transport velocity of the bubble is expressed as

$$V \sim \frac{\Delta T^2}{(\mu_0 + \mu_1)} \left[D \left(\frac{3\pi^2}{48Ra^2} - \frac{4\pi^4 + \pi^2}{192R^3} \right) + L \right] \quad \text{R(7)}$$

Where $D = -\pi\epsilon_0(\epsilon_A - \epsilon_W)K_e^2 P_e^2 A^2$ and $L = dP_e^2 / (\epsilon_0 \epsilon_a)$ are two constants, ϵ_A and ϵ_W are the relative dielectric constant of air, and the relative dielectric constant of water, respectively, A is the area of the NIR spot, a is the distance of the bubble edge from the spot center, K_e is the Coulomb constant, μ_0 and μ_1 are the viscosity of oil and liquid, respectively and ΔT is the temperature rise of the lithium niobate crystal at illumination. To obtain the temperature rise ΔT under NIR irradiation, we first consider the heat transfer in the photothermal layer, which can be simplified to be heat conduction in a semi-infinite cylinder, and described as¹⁰

$$\frac{1}{b} \frac{\partial T}{\partial t} = \frac{1}{r} \frac{\partial(r \frac{\partial T}{\partial r})}{\partial x}, \quad 0 < r < \infty \quad \text{R(8)}$$

$$t = 0, \quad T(r, t) = T_0 \quad \text{R(9)}$$

$$r = R_0, \quad -\lambda \frac{\partial T}{\partial r} = q_0 \quad \text{R(10)}$$

where b , t , r , T_0 , λ and q_0 are the thermal diffusivity of photothermal film, irradiation time, the distance from any point on the surface to the laser spot center, ambient temperature, thermal conductivity and heat flux, respectively. Here, the heat flux can

be expressed as $q_0=P/(2\pi R_0d_0)$, where P and d_0 are the laser power and photothermal layer thickness, respectively. The analytical solution of the problem can be readily obtained as

$$T(r, t) = T_0 + \frac{P}{\lambda\pi^2 d_0} I \quad (\text{R11})$$

$$I = \int_0^\infty \frac{1-e^{-\frac{btu}{R_0}}}{u^2} \times \frac{Y_1(u)J_0\left(\frac{ru}{R_0}\right)-J_1(u)Y_0\left(\frac{ru}{R_0}\right)}{J_1^2(u)+Y_1^2(u)} du \quad (\text{R12})$$

Assuming the same temperature for the photothermal layer and the lithium niobate due to their intimate contact, the temperature rise of lithium niobate crystal at the irradiation point can be expressed as

$$\Delta T(r, t) = T(r, t) - T_0 = \frac{P}{\lambda\pi^2 d_0} I \quad (\text{R13})$$

Therefore, the temperature rise at the edge of the bubble under NIR irradiation is

$$\Delta T(R_0, t) = T(R_0, t) - T_0 = \frac{P}{\lambda\pi^2 d_0} I(R_0, t) \quad (\text{R14})$$

$$I(R_0, t) = \int_0^\infty \frac{1-e^{-\frac{btu}{R_0}}}{u^2} \times \frac{Y_1(u)J_0(u)-J_1(u)Y_0(u)}{J_1^2(u)+Y_1^2(u)} du \quad (\text{R15})$$

A combination of Equation (R7) and Equation (R14) leads to

$$V \sim \frac{P^2 I^2(R_0, t)}{\lambda^2 \pi^4 d_0^2 (\mu_o + \mu_i)} \left[D \left(\frac{3\pi^2}{48Ra^2} - \frac{4\pi^4 + \pi^2}{192R^3} \right) + L \right] \quad (\text{R16})$$

Equation (R16) clearly shows that the bubble transport velocity is related to the laser power P , laser spot size R_0 and the bubble size R . Therefore, for bubbles with different sizes, a simple relation between the bubble transport velocity and the relative size of the area illuminated by the laser respect to the bubble size, i.e., R_0/R , is not available. We have added this information in the revised Supplementary Materials as highlighted in red.

Fig. R6 | Bubble manipulation on the PESS with a volume of 1 μL (a) and 70 μL (b), respectively.

(2) The scalability of the system is reflected in the following aspects. First, versatile bubble maneuvering on photopyroelectric slippery surfaces can be achieved within a

wide volume range, i.e., 1 to 70 μl , as shown in Fig. R6. Moreover, multiple bubbles can be controlled in a synchronous manner, as shown in Fig. R7. In addition to bubble maneuvering, this design paradigm is capable of manipulating droplets in air and oil, as shown in Fig. R8.

Fig. R7 | Sequential images showing the manipulation of multiple bubbles on the PESS. **a** Two bubbles move to the right simultaneously under NIR irradiation. **b** Three bubbles were driven away from the light source.

Fig. R8 | Droplet maneuvering on the PESS under NIR irradiation in air (**a**) and oil (**b**), respectively.

(3) In order to verify whether the surface tension destroys the scalability of the system, further bubble manipulation by changing the surface tension of the surrounding liquid or the lubricating oil is presented. Note that the difference of the surface tension for commonly used lubricants such as silicone oil, perfluoropolyether and mineral oil is too small to reflect their influence on the system, and therefore, we mainly focus on the varied surface tension of the surrounding liquid. It is obvious from Fig. R9a that the bubble volume accurately manipulated gets smaller when the surface tension of the surrounding liquid is reduced, which indicates that the surface tension holds a significant impact on the scalability of the system.

Fig. R9 | Effect of the surface tension of the surrounding liquid on the system scalability. **a** Bubble volume accurately manipulated as a function of the liquid surface tension. **b** Schematics showing the forces on the bubble in the vertical direction. **c** Schematic showing the replacement of the lubricant oil on the slippery layer by the surrounding liquid with a low surface tension.

Specifically, two main approaches affecting the system scalability by changing the surface tension were identified in the experiment. First, the vertical component of the capillary force $F_{CV} = 2\pi\gamma_{LG}R \sin \theta$ exerting on the bubble reduced with a decreased surface tension of the surrounding liquid, enabling the detachment of the bubble from the surface directly due to the excess of the buoyancy force F_B over F_{CV} (Fig. R9b). Therefore, it becomes more difficult to control bubbles with a large size, which tend to float to the surface directly (Bubble 1 and 2 in Fig. R9a). Moreover, the surrounding liquid with a surface tension comparable to or even smaller than that of the lubricating oil will easily replace the lubricating oil (Fig. R9c), engendering the lubrication failure of the surface,^{11,12} followed by the pinning of the bubbles (Bubble 3 and 4 in Fig. R9a).

3-Following the previous question. The authors should clarify why the transport in a closer system is harder than in an open one. Is it due to the viscous friction with the upper plate? How does the friction scales with the bubble size? Would be possible to use semi-transparent PESS to induce a force in both sides of the bubble? (i.e. top and bottom). Closely related, the mechanisms of bubble splitting needs also further explanation.

Response: We thank the reviewer for pointing out this issue. There are several points

embedded in this comment, for which we would like to address one by one.

Fig. R10 | Schematics showing the bubble shape in an open (a) and closed (b) system, respectively.

(1) We are sorry for the misunderstanding due to our unclear description. When we say the resistance in a closed system is larger, what we mean is that the adhesion force of the bubble, i.e., the force required to accelerate the bubble from a static state to a dynamic state between two surfaces is larger, but not the hydrodynamic resistance force when it slides. The adhesive force of a bubble in an open (F_{R1}) and closed (F_{R2}) system can be respectively expressed as¹³⁻¹⁶

$$F_{R1} = \gamma_{LG}\pi R(\cos \theta_r - \cos \theta_a) = \rho g \Omega \sin \theta_{s1} \quad (R17)$$

$$F_{R2} = \gamma_{LG}\pi R_1(\cos \theta_{r1} - \cos \theta_{a1}) = \rho g \Omega \sin \theta_{s2} \quad (R18)$$

where R_1 is the contact radius of the bubble with the surface in a closed system (Fig. R10), θ_r , θ_a , θ_{r1} and θ_{a1} are the receding and advancing angles of the bubble in an open and closed system, respectively. ρ , Ω and g are the water density, bubble volume and acceleration of gravity, respectively. θ_{s1} and θ_{s2} represent the sliding angles of a bubble in an open and closed system, respectively. As shown in Fig. R11, for a bubble with a fixed volume, a smaller sliding angle in the open system indicates a lower adhesive force than that in the closed system. Moreover, a more compressed bubble, i.e., $d_h=1$ mm, is more difficult to accelerate from a static state.

Fig. R11 | The sliding angle of a bubble as a function of its volume in both the open and closed system from the experiments.

As a bubble moves forward, a backward hydrodynamic resistance force comprising lubricant and liquid media's viscous forces is generated both in the open and closed system, which could be respectively expressed as^{9,17,18}

$$F_{H1} = \alpha_1 \pi V R (\mu_o + \mu_l) \quad (R19)$$

$$F_{H2} = \alpha_2 \pi V \left(\frac{2\mu_l R_l^2}{d_h} + 2R_l \mu_o \right) \quad (R20)$$

where α_1 and α_2 are numerical factors. Equations (R19) and (R20) clearly show that the hydrodynamic resistance of the bubble is related to various factors including the numerical factor, bubble sliding speed, the distance between the parallel plates, and etc. Therefore, more detailed investigations are required in order to determine the force value in the two systems. We will take the reviewer's suggestions to further investigate the influence factors their values on the hydrodynamic resistance in our future study.

We have added more information to make this point clear in the revised manuscript as highlighted in red: "Usually, bubble transport in a closed system is more challenging than that in an open one due to the generated lager **adhesion** force between the upper and lower plates **when accelerating a bubble from a static state to a dynamic one**⁴⁵".

Figure R12 | The adhesive force as a function of the bubble volume both in the open and closed system.

(2) If we understand it correctly, the reviewer wonders the adhesion force of the bubble, i.e., the force required to accelerate the bubble from a static state to a dynamic state, and the hydrodynamic resistance force during the transport of bubbles with different size. For the former, because the adhesion force is determined by both the bubble volume and the sliding angle, no explicit variation tendency of the force with the volume is available, as shown in Fig. R12. For the latter, Equations (R19) and (R20)

clearly show that the hydrodynamic resistance force is positively related to the bubble size both in an open system and a fixed closed system.

(3) Following the reviewer's advice, we have fabricated a semi-transparent PESS by reducing the content of Fe_3O_4 in the photothermal layer to enable the NIR irradiation on both surfaces, i.e., the NIR light permeates through a semi-transparent PESS and reaches a standard PESS. Similarly, the bubble was driven away from the light source under NIR irradiation (Fig. R13a).

Fig. R13 | Sequential images showing the transport of bubbles compressed between a semi-transparent PESS and a standard PESS from the top view (**a**) and side view (**b**), respectively.

To verify the driving force generation on both sides, we examined the shape change of the bubble before and after NIR irradiation from the side view. As shown in Fig. R13b, the contact angles between the bubble and upper and lower surfaces both increased, indicating the generation of dielectric wetting and therefore the driving force on both surfaces for the bubble transport. Note that less energy is absorbed, leading to a lower-efficiency bubble transport, due to the energy degradation after the light passing through the semi-transparent PESS.

(4) Mechanism of bubble splitting on photopyroelectric slippery surfaces

Bubble splitting between two parallel plates. As shown in Fig. R14a, when compressed by a parallel plate, a bubble on the PESS forms a concave surface, which is subjected to a uniform outward Laplace force in the radial direction. Note that, the water contact angles at the upper (θ_{a0}) and lower (θ_0) surface are equal to each other, i.e., $\theta_{a0} = \theta_0$. When the NIR light irradiating left side of the bubble, as shown in Fig. R14b, the water contact angle θ on the bottom surface decreases due to the generated dielectric wetting⁶, which could be expressed as

$$\cos \theta_1 = \cos \theta_{10} + \frac{\varepsilon_0 \varepsilon_d}{2\gamma_{LG} d} \Delta V^2 = \cos \theta_{10} + \frac{P_C^2 \Delta T^2 d}{2\gamma_{LG} \varepsilon_0 \varepsilon_d} \quad (\text{R21})$$

Fig. R14 | Experimental image and schematics showing the mechanism of bubble splitting between two parallel plates without (a) and with (b) NIR irradiation.

Obviously, after switching on the NIR light, a Laplace pressure towards the inside of the bubble generates at the left side due to the transformation of the concave meniscus (Fig. R14a) to a convex one (Fig. R14b) to split the bubble, which can be expressed as

$$P_0 = \gamma_{LG} \left(\frac{1}{r_1} + \frac{1}{r_2} \right) \quad (\text{R22})$$

The radius of the meniscus of the bubble r_2 is equal to the radius of NIR spot $R_0 = 0.5$ mm. The radius of the meniscus of the bubble r_1 is geometrically related to the local contact angles and the distance between the plates d_h , which could be expressed as

$$r_1 = \frac{d_h}{\cos \theta_u + \cos \theta_l} \quad (\text{R23})$$

where θ_u is the contact angle of the water with the upper surface after NIR irradiation. A combination of Equations (R22) and (R23) leads to the Laplace pressure to split the bubble as

$$P_0 = \gamma_{LG} \left(\frac{\cos \theta_u + \cos \theta_l}{d_h} + \frac{1}{R_0} \right) \quad (\text{R24})$$

It is clear from Equation (R24) that the Laplace pressure for bubble splitting increased after reducing the distance between two plates d_h , which is in line with the experimental results. Notably, the dielectrophoretic force shows little effect on the bubble splitting because it acts on the whole bubble^{7,8}.

Fig. R15 | Schematics showing bubble splitting process on the PESS without (a) and with (b) NIR irradiation.

Bubble splitting between two inclined plates. When a bubble is compressed by an inclined plate on the PESS, a Laplace force is generated to drive the bubble to the tip⁹ (Fig. R15a), and under NIR irradiation, the bubble deforms to generate a Laplace pressure towards the inside of the bubble to splitting the bubble (Fig. R15b), which can be calculated by

$$P_0 = \gamma_{LG} \left(\frac{\cos \theta_u + \cos \theta_l}{D_1 \tan \beta} + \frac{1}{R_0} \right) \quad (\text{R25})$$

It is clear that the Laplace pressure for bubble splitting increases at a reduced β , which is in line with the experimental result. We have added this information in the revised manuscript as highlighted in red: “...engenders an arbitrary bubble segmentation **due to the Laplace pressure towards the interior of the bubble caused by dielectric wetting (Supplementary Discussion IV).**”.

4-It would be interesting that the authors further discuss the origin of the discrepancy between the expected saturation voltage (V_s) estimated by and the one verified by the experiments. This also applies to the saturation temperature. As the whole phenomenon is thermally driven I think that this might be a relevant issue to explain.

Response: Following the reviewer’s comment, we conducted experiment to measure the saturation temperature change ΔT_s , and then the saturation voltage V_s could be calculated according to $V_s = \frac{P_c \Delta T_s d}{\epsilon_0 \epsilon_d}$, with P_c , d , ϵ_0 and ϵ_d being the pyroelectric coefficient of the pyroelectric crystal, the thickness of the dielectric layer, the permittivity of vacuum and the relative dielectric constant of the dielectric layer,

respectively. Notably, $\Delta T_s \approx 7.2$ °C is obtained from the experiment, corresponding to a saturation voltage $V_s \approx 134.9$ V, which is slightly smaller than the theoretical one as 144.3 V under a saturation temperature change $\Delta T = 7.7$ °C. This discrepancy is attributed to the fact that the temperature rise was not taken into account in calculating the saturation voltage in traditional dielectric wetting, which is expressed as

$$V_s = [2d\gamma_{OL}/\epsilon_0\epsilon_d]^{0.5} = [2d(\gamma_{OG} - \gamma_{LG}\cos\theta_0)/\epsilon_0\epsilon_d]^{0.5} \quad (\text{R26})$$

In our research, continuous heating is involved due to the photothermal effect under NIR irradiation, and therefore a reduced oil-gas surface tension γ_{OG} and θ_0 (θ_0 is the contact angle without external voltage¹⁹) is obtained, leading to the decrease of V_s . This is consistent with the above experimental observation and the result in previous research²⁰. However, the formula can be still used to predict the saturation voltage due to the small temperature rise.

Fig. R16 | Schematic showing the structure of the PESS. As the NIR light irradiates, the photothermal layer produces heat because of photothermal effect. The temperature within the pyroelectric crystal rises synchronously due to thermal conduction, giving rise to extra surface free charges, which drives the bubble into motion.

5-It would be helpful for the reader to include the direction of the gravitation acceleration in Figure 1 (a) to make clear that the bubble is sitting on the PESS.

Response: We thank the referee for his/her comment on the improvement of clarity in our presentation. In line with the referee’s comment, we have added the direction of the gravitation acceleration in the revised Figure 1 (a) to make clear that the bubble is sitting on the PESS, as shown in Fig. R16.

6-In the caption of Figure 1 (b), the authors could clarify that the thermal image is a picture and not a COMSOL simulation.

Response: We thank the reviewer very much for her/his suggestion on improving the presentation of our finding. We have clarified this by modifying the caption of Figure 1(a) as “The inset shows the infrared thermal captured by an infrared camera at ~ 80 s.”.

7-Are there any permanent changes as a consequence of the repeated localized heating and cooling cycles applied to the PESS? e.g. silicon evaporation or porous structure cracks produced by thermal stresses.

Response: Thank the reviewer for this insightful comment. Following the reviewer’s advice, the durability of the PESS is verified by repeated in-situ heating and cooling experiments, during which no obvious morphology and mass change of the surface is observed (Fig. R17) after 300 cycles of impulse NIR irradiation with a power of 500 mW at a frequency of 0.5-s on and 5-s off at room temperature.

Fig. R17 | Durability verification of the PESS after 300 cycles of impulse NIR irradiation. **a** Surface morphology of the PESS at the 1st, 100th, 200th and 300th heating and cooling cycles. **b** Mass change of PESS during 300 repeated local heating and cooling cycles.

8-A bubble velocity of 4.5 mm s⁻¹ described as “ultrafast” is questionable. Previous works with drops reports more than 33 mm s⁻¹.

Response: We thank the reviewer for bringing up this point into our attention. We agree with the reviewer that previous study² has shown that the transport velocity of a droplet under NIR irradiation can exceed 33 mm s^{-1} . However, the photothermal conversion efficiency of an aqueous environment for bubble manipulation is much lower than that for a droplet in air. Moreover, the hydrodynamic resistance for a bubble transport in a liquid media is much larger compared to the droplet motion in air. Therefore, the bubble moving velocity in a liquid environment is usually much slower than that of a droplet in air, and a speed of 4.5 mm s^{-1} can be considered remarkable in comparison to other underwater bubble transmissions. In order to avoid any misunderstanding, we have changed the word from “reach an ultrafast velocity over 4.5 mm s^{-1} on the PESS” to “reach a velocity over 4.5 mm s^{-1} on the PESS” in the revised manuscript.

9-“In general, a larger bubble requires a lower laser power to enable an easier bubble escape and vice versa.” This should be explained not only on the Supplemental material.

Response: We thank the reviewer for her/his comment on the improvement of clarity in our presentation. In line with the reviewer’s comment, we have added more information to explain this point clearly in the revised manuscript as highlighted in red: “In general, a larger bubble necessitates a lower laser power to enable an easier bubble escape and vice versa (Supplementary Fig. 15). This is because larger bubbles are subjected to a greater upward buoyancy force, and correspondingly requiring a smaller additional upward dielectrophoretic force by a lower power laser to overcome the downward capillary force for a bubble detachment (Supplementary Discussion V).”.

10-In the section dedicated to “microrobot” the authors state all the benefits of the technique but they omitted to mention how the bubbles could be introduced in the system on the first place.

Response: We thank the reviewer for bringing up this point into our attention. Currently, there are two methods for introducing bubbles into the system. One is to directly inject bubbles using an injector, which possess the advantages of simple and fast operation, good accuracy but can cause interference to the system. The second one is generating surface bubbles in water by leveraging the photothermal effect²¹. Specifically, dispersed localized photothermal zones can be produced from the mixture of Fe_3O_4 nanoparticles

and PDMS, which is sandwiched between the slippery layer and the pyroelectric layer during the preparation process of the PESS (Fig. R18a). When the photothermal zone is irradiated by the NIR light, part of the light energy is converted into heat energy and thus a bubble is generated due to the photothermal effect (Fig. R18b). Note that, this method is time-consuming and difficult to accurately control the bubble size. We have added this information to the Methods part in the revised manuscript as highlighted in red: “Two methods were proposed for introducing bubbles into the bubble manipulation system. One is to directly inject bubbles using an injector, where the bubble size can be accurately controlled. The second one is generating surface bubbles in water by leveraging the photothermal effect³³ (Supplementary Fig. 16), which is only used for underwater salvage in the experiment.”.

Fig. R18 | Surface bubble generation by leveraging the photothermal effect. **a** Schematic showing the mechanism for surface bubble generation. **b** Sequential images showing the generation of a surface bubble under NIR irradiation.

11-What is the beam profile of the laser? What is the laser power and precise wavelength? Model?

Response: We thank the reviewer for bringing up this point into our attention. We used a near-infrared laser with a wavelength of 808nm and a Gaussian beam profile. The laser power types available were 100mw, 200mw, 300mw, 500mw, and 1000mw, corresponding to the models FU808AD100-16GD, FU808AD200-16GD, FU808AD300-16GD, FU808AD500-16GD, and FU808AD1000-16GD, respectively.

We have added this information in the Methods part in the revised manuscript as highlighted in red: “The near infrared laser with a spot size 1×1 mm is provided by Shenzhen Fuzhe Technology Co., Ltd., China. Specifically, 808-nm NIR lasers in a Gaussian beam profile with different powers of 100 mw, 200 mw, 300 mw, 500 mw, and 1000 mw were used for the bubble manipulation, corresponding to the laser models of FU808AD100-16GD, FU808AD200-16GD, FU808AD300-16GD, FU808AD500-16GD, and FU808AD1000-16GD, respectively.”.

Reviewer #2 (Remarks to the Author):

Report on the paper by Zhan et al.

It is an excellent paper on versatile bubble maneuvering on photopyroelectric slippery surfaces. However, there are some unclear points.

Response: We really appreciate the reviewer's positive feedback and the time she/he has taken to make constructive comments on our work. In the following, we respond point by point to each of the reviewer's comments.

1. Line 32-34, "For instance, inspired by cactus spines and pitcher plants, slippery coper cones with geometry gradient were developed to achieve the gas bubble transportation." It is required to cite some references for this statement.

Response: We thank the reviewer for bringing up this point into our attention. According to the reviewer's suggestion, the following three references have been cited appropriately in the revised manuscript as highlighted in red: "For instance, inspired by cactus spines and pitcher plants, slippery coper cones with geometry gradient were developed to achieve the gas bubble transportation.¹³⁻¹⁵".

[13] Xiao, X. et al. Bioinspired slippery cone for controllable manipulation of gas bubbles in low-surface-tension environment. *ACS Nano* **13**, 4083-4093 (2019).

[14] Shi, D. et al. Ladderlike conical micropillars facilitating underwater gas-bubble manipulation in an aqueous environment. *ACS Appl. Mater. Interfaces* **12**, 42437-42445 (2020).

[15] Zhang, C. et al. Bioinspired pressure-tolerant asymmetric slippery surface for continuous self-transport of gas bubbles in aqueous environment. *ACS Nano* **12**, 2048-2055 (2018).

2. With regard to the manipulation on a photothermal slippery surface (PSS) shown in Supplementary Fig. 5, it is required to explain the mechanism of the manipulation in the manuscript.

Response: We thank the reviewer for bringing up this point into our attention. As reported previously⁵, the bubble transportation on photothermal slippery surfaces under NIR irradiation is attributed to the Laplace force generated by the NIR-induced asymmetric deformation of bubbles. Initially, the left and right contact angles of the bubble on the PSS without irradiation are equal to each other as $\theta_{a0}=\theta_{b0}=\theta$ (Fig. R19), which can be expressed by

$$\cos \theta = \frac{\gamma_{OL}-\gamma_{OG}}{\gamma_{LG}} \quad (R27)$$

where γ_{OG} , γ_{OL} and γ_{LG} are the tensions of the oil-gas, oil-liquid and liquid-gas interfaces, respectively. When the NIR light irradiating on the left side of the bubble, the temperature of the illuminated position rises quickly, which leads to a decreased γ_{OG} , corresponding to a decreased θ_{a0} while θ_{b0} remains relatively unchanged (Fig. R19) and a asymmetric deformation of the bubble. This asymmetric deformation of the bubble generates a Laplace force $F_L=2R\gamma_{LG}(\cos \theta_{a1} - \cos \theta_{b0})$ that drives the bubble to chase the light source. By contrast, the surface tension change can be neglected on the PESS due to due to the low temperature rise. We have added this information in the revised manuscript as highlighted in red: “which is directly opposite to that on a photothermal slippery surface (PSS), where a bubble atop the surface could be driven toward the irradiated spot **due to the generated Laplace force by the decrease of the bubble contact angle under NIR irradiation** (Supplementary Fig. 5 and Supplementary Discussion I).”.

Fig. R19 | Schematics showing the force analysis in NIR-induced bubble transport on the photothermal slippery surface. F_H is the resistance force.

3. It is required to explain in the manuscript why zeta potential of a bubble is not taken into account in eq. (1).

Response: We thank the reviewer for this insightful comment. The zeta potential of air bubble is formally the electrostatic potential at the ‘plane of slip’²², which is measured to be ~ -65 mV at the air/water interface²³. Whether or not zeta potential of a bubble should be taken into consideration depends on its impact on the change of the air/water interfacial tension in the system^{24,25}. Various studies have clearly indicated that the applied electric field from a High Voltage Supplier, e.g., with a value in the order of 100 V, has insignificant effect on the surface tension in the air/water interface^{26,27}. Therefore, zeta potential of the bubble could be omitted in Young-Lippmann equation²⁸⁻³⁰. We have added this information in the revised manuscript as highlighted in red: “Note that the zeta potential of bubbles is not taken into account in in Equation (1), due to its negligible effect on dielectric wetting³⁶⁻³⁸.”.

4. With regard to eq. (S10), dimension of $F R^{-1}$ is kg s^{-2} which differs from the dimension of velocity m s^{-1} . Thus eq. (S10) seems to be wrong.

Response: Thank the reviewer for pointing out our mistakes. In this research, a moving bubble is mainly subjected to two driving forces, Laplace force (F_L), dielectrophoretic force (F_{DX}), and a viscous resistance ($F_H \approx \alpha \pi V R (\mu_o + \mu_l)$). Therefore, the resultant force driving the bubble under NIR irradiation on the PESS is $F = F_L + F_{DX} - F_H$. Obviously, a stationary bubble starts to accelerate after switching on the NIR light. F_H continues to increase as the bubble speeds up until a balance between the driving force and resistance force ($F_L + F_{DX} = F_H$), at which point the bubble reaches its steady velocity scaled as $V \sim (F_L + F_{DX}) / [R(\mu_o + \mu_l)]$ but not the one ($V \sim (F_L + F_{DX}) / R$) in the last version. Where R is the radius of the bubble, μ_o and μ_l are the viscosity of oil and liquid, respectively. In our system, $\mu_o + \mu_l$ is a constant. Obviously, The dimension of $(F_L + F_{DX}) / [R(\mu_o + \mu_l)]$ is $\text{N} / (\text{m Pa}\cdot\text{s}) = \text{N} / [\text{m}\cdot(\text{N}/\text{m}^2)\cdot\text{s}] = \text{m}/\text{s}$, which is consistent with the dimension of velocity. Now we have revised the formula in the revised manuscript and Supplementary Materials as highlighted in red: “at which point the bubble reaches its steady velocity scaled as $V \sim (F_L + F_{DX}) / [R(\mu_o + \mu_l)]$ ”.

Reference

1. Wang, F. et al. Light-induced charged slippery surfaces. *Sci. Adv.* **8**, eabp9369 (2022).
2. Wang, F. et al. Light control of droplets on photo-induced charged surfaces. *Natl. Sci. Rev.* **10**, nwac164 (2023).
3. Munoz-Martinez, J. F. et al. Diffractive optical devices produced by light-assisted trapping of nanoparticles. *Opt. Lett.* **41**, 432-435 (2016).
4. Sebastián-Vicente, C., Muñoz-Cortés, E., García-Cabañas, A., Agulló-López, F. & Carrascosa, M. Real-time operation of photovoltaic optoelectronic tweezers: New strategies for massive nano-object manipulation and reconfigurable patterning. *Part. Part. Syst. Charact.* **36**, 1900233 (2019).
5. Chen, C. et al. Remote photothermal actuation of underwater bubble toward arbitrary direction on planar slippery Fe₃O₄-doped surfaces. *Adv. Funct. Mater.* **29**, 1904766 (2019).
6. Guan, Y. & Tong, A. Y. A numerical study of microfluidic droplet transport in a parallel-plate electrowetting-on-dielectric (EWOD) device. *Microfluid. Nanofluid.* **19**, 1477-1495 (2015).
7. Jones, T. B. Dielectrophoretic force calculation. *J. Electrostat.* **6**, 69-82 (1979).
8. Zhao, Y. & Cho, S. K. Micro air bubble manipulation by electrowetting on dielectric (EWOD): transporting, splitting, merging and eliminating of bubbles. *Lab Chip* **7**, 273-280 (2007).
9. Prakash, M., Quere, D. & Bush, J. W. Surface tension transport of prey by feeding shorebirds: the capillary ratchet. *Science* **320**, 931-934 (2008).
10. Li, W., Tang, X. & Wang, L. Photopyroelectric microfluidics. *Sci. Adv.* **6**, eabc1693 (2020).
11. Preston, D. J., Song, Y., Lu, Z., Antao, D. S. & Wang, E. N. Design of lubricant infused surfaces. *ACS Appl. Mater. Interfaces* **9**, 42383-42392 (2017).
12. Wong, T. S. et al. Bioinspired self-repairing slippery surfaces with pressure-stable omniphobicity. *Nature* **477**, 443-447 (2011).
13. Zhuang, K., Lu, Y., Wang, X. & Yang, X. Architecture-driven fast droplet transport without mass loss. *Langmuir* **37**, 12519-12528 (2021).
14. Lin, F., Wo, K., Fan, X., Wang, W. & Zou, J. Directional transport of underwater bubbles on solid substrates: Principles and applications. *ACS Appl. Mater. Interfaces* **15**, 10325-10340 (2023).
15. Zhuang, K., Yang, X., Huang, W., Dai, Q. & Wang, X. Efficient bubble transport on bioinspired topological ultraslippery surfaces. *ACS Appl. Mater. Interfaces* **13**, 61780-61788 (2021).
16. Zhu, S. et al. Spontaneous and unidirectional transportation of underwater bubbles on superhydrophobic dual rails. *Appl. Phys. Lett.* **116**, 093706 (2020).
17. Bahadur, V. & Garimella, S. V. An energy-based model for electrowetting-induced droplet

- actuation. *J. Micromech. Microeng.* **16**, 1494-1503 (2006).
18. Bjelobrk, N. et al. Thermocapillary motion on lubricant-impregnated surfaces. *Phys. Rev. Fluids* **1**,063902 (2016).
 19. Quinn, A., Sedev, R. & Ralston, J. Contact angle saturation in electrowetting. *J. Phys. Chem. B* **109**, 6268-6275 (2005).
 20. Liu, J., Wang, M., Chen, S. & Robbins, M. O. Uncovering molecular mechanisms of electrowetting and saturation with simulations. *Phys. Rev. Lett.* **108**, 216101, (2012).
 21. Dai, L., Ge, Z., Jiao, N. & Liu, L. 2D to 3D manipulation and assembly of microstructures using optothermally generated surface bubble microrobots. *Small* **15**, e1902815, (2019).
 22. Karraker, K. A. & Radke, C. J. Disjoining pressures, zeta potentials and surface tensions of aqueous non-ionic surfactant/electrolyte solutions: theory and comparison to experiment. *Adv. Colloid Interface Sci.* **96**, 231-264 (2002).
 23. Graciaa, A., Morel, G., Saulner, P., Lachaise, J. & Schechter, R. S. The ζ -potential of gas bubbles. *J. Colloid Interface Sci.* **172**, 131-136 (1995).
 24. Orejon, D., Sefiane, K. & Shanahan, M. E. R. Young-Lippmann equation revisited for nano-suspensions. *Appl. Phys. Lett.* **102**, 201601 (2013).
 25. Hu, X. et al. On the performance of thermostable electrowetting agents. *Surf. Interface Anal.* **44**, 478-483 (2012).
 26. Hayes, C. F. Water-air interface in the presence of an applied electric field. *J. Phys. Chem.* **79**, 1689-1693 (2002).
 27. Jiang, Q., Chiew, Y. C. & Valentine, J. E. Electric field effects on the surface tension of air/solution interfaces. *Colloids Surf. A Physicochem. Eng. Asp.* **83**, 161-166 (1994).
 28. Yan, R., Pham, R. & Chen, C. L. Activating bubble's escape, coalescence, and departure under an electric field effect. *Langmuir* **36**, 15558-15571 (2020).
 29. Yan, Y. et al. Electrowetting-induced stiction switch of a microstructured wire surface for unidirectional droplet and bubble motion. *Adv. Funct. Mater.* **28**, 1800775 (2018).
 30. Arscott, S. Electrowetting of soap bubbles. *Appl. Phys. Lett.* **103**, 014103 (2013).

REVIEWERS' COMMENTS

Reviewer #1 (Remarks to the Author):

The authors have satisfactorily addressed all of my comments and criticisms, providing comprehensive explanations and improving both the manuscript's text and figures. Therefore, I can now recommend the manuscript for publication.

Juan Manuel Rosselló

Reviewer #2 (Remarks to the Author):

As the comments have been addressed in the revised manuscript, I would like to recommend the publication of the paper.

Point-to-point response to reviewers' comments

Reviewer #1 (Remarks to the Author):

The authors have satisfactorily addressed all of my comments and criticisms, providing comprehensive explanations and improving both the manuscript's text and figures. Therefore, I can now recommend the manuscript for publication.

Juan Manuel Rosselló

Response: We thank Prof. Juan Manuel Rosselló for his recommendation for publication of our work and the time he has taken to make constructive comments on the manuscript, which definitely helped to improve its quality.

Reviewer #2 (Remarks to the Author):

As the comments have been addressed in the revised manuscript, I would like to recommend the publication of the paper.

Response: We thank the anonymous reviewer again for her/his recommendation for publication of our work and the time she/he has taken to make constructive comments on the manuscript, which definitely helped to improve its quality.